# Learning Symmetric Collaborative Dialogue Agents with Dynamic Knowledge Graph Embeddings

## Abstract

We study a *symmetric collaborative dialogue* setting in which two agents, each with private knowledge, must strategically communicate to achieve a common goal. The open-ended dialogue state in this setting poses new challenges for existing dialogue systems. We collected a dataset of 11K human-human dialogues, which exhibits interesting lexical, semantic, and strategic elements. To model both structured knowledge and unstructured language, we propose a neural model with dynamic knowledge graph embeddings that evolve as the dialogue progresses. Automatic and human evaluations show that our model is both more effective at achieving the goal and more human-like than baseline neural and rule-based models.

## 1 Introduction

Current task-oriented dialogue systems (Young et al., 2013; Wen et al., 2017; Dhingra et al., 2016) require a pre-defined dialogue state (e.g., slots such as food type and price range for a restaurant searching task) and a fixed set of dialogue acts (e.g., request, inform). However, human conversation often requires richer dialogue states and more nuanced, pragmatic dialogue acts. In contrast, recent open-domain chat systems (Shang et al., 2015; Serban et al., 2015b; Sordoni et al., 2015; Li et al., 2016a) learn a mapping directly from previous utterances to the next utterance. While these models capture open-ended aspects of dialogue, the lack of structured dialogue state prevents them from being directly applied to settings that require interfacing with structured knowledge.

In order to bridge the gap between the two types of systems, we focus on a *symmetric collabora-*

Friends of agent A:

| Name | School | Major | Company |
|------|--------|-------|---------|
| Jessica | Columbia | Computer Science | Google |
| Josh | Columbia | Linguistics | Google |
| ... | ... | ... | ... |

A:  Hi! Most of my friends work for Google
B:  do you have anyone who went to columbia?
A:  *Hello?*
A:  I have Jessica a friend of mine
A:  and Josh, both went to columbia
B:  *or anyone working at apple?*
B:  SELECT (Jessica, Columbia, Computer Science, Google)
A:  SELECT (Jessica, Columbia, Computer Science, Google)

Figure 1: An example dialogue from the Mutual-Friends task, in which two agents, A and B, each given a private list of a friends, try to identify their mutual friend. Our objective is to build an agent that can perform the task with a human. Cross-talk (Section 2.3) is *italicized*.

*tive dialogue* setting, which is task-oriented but encourages open-ended dialogue acts. In our setting, two agents, each with a private list of items with attributes, must communicate to identify the unique shared item. Consider the dialogue in Figure 1, in which two people are trying to find their mutual friend. When B asks "do you have anyone who went to columbia?", it suggests that she has some Columbia friends and they probably work at Google. Such conversational implicature is lost when interpreting it as simply requesting information about Columbia. In addition, it is hard to define a state that captures the diverse semantics in these utterances (e.g., defining "most of", "might be"; see details in Table 1).

To model both structured and open-ended context, we propose the *Dynamic Knowledge Graph Network* (DynoNet), in which the dialogue state is modeled as a knowledge graph with an embedding for each node (Section 3). Our model is similar

to EntNet (Henaff et al., 2017) in that node/entity embeddings are updated recurrently given new utterances. The difference is that we structure entities as a knowledge graph; as the dialogue proceeds, new nodes are added and new context is propagated on the graph. An attention-based mechanism (Bahdanau et al., 2015) over the node embeddings drives generation of new utterances. Our model's use of knowledge graphs captures the grounding capability of classic task-oriented systems and the graph embedding provides the representational flexibility of neural models.

The naturalness of communication in the symmetric collaborative setting enables large-scale data collection: We were able to crowd-source around 11K human-human dialogues on Amazon Mechanical Turk (AMT) in less than 15 hours.[1] We show that the new dataset calls for more powerful representation beyond fully-structured states (Section 2.2).

Aside from the third-party human evaluation adopted by most work (Liu et al., 2016; Li et al., 2016b,c), we also conduct partner evaluation (Wen et al., 2017) where AMT workers rate their conversational partners (our models) based on fluency, correctness, cooperation, and human-likeness. We compare DynoNet with baseline neural models and a strong rule-based system. The results show that DynoNet can perform the task with humans efficiently and naturally; it also captures some strategic aspects of human-human dialogues.

The contributions of this work are: (i) a new symmetric collaborative dialogue setting and a large dialogue corpus that push the boundaries of existing dialogue systems; (ii) DynoNet that integrates semantically rich utterances with structured knowledge to represent open-ended dialogue states; (iii) multiple automatic metrics based on bot-bot chat and a comparison of third-party and partner evaluation.

## 2 Symmetric Collaborative Dialogue

We introduce a collaborative task between two agents and describe the human-human dialogue collection process. We show that our data exhibits diverse, interesting language phenomena.

### 2.1 Task Definition

In the symmetric collaborative dialogue setting, there are two agents, A and B, each with a private knowledge base—$KB_A$ and $KB_B$, respectively. Each knowledge base includes a list of *items*, where each item has a value for each *attribute*. For example, in the MutualFriends setting, Figure 1, items are friends and attributes are name, school, etc. There is a shared item that A and B both have; their goal is to converse with each other to determine the shared item and select it. Formally, an agent is a mapping from its private KB and the dialogue thus far (sequence of utterances) to the next utterance to generate. A dialogue is considered *successful* when both agents correctly select the shared item. This setting has parallels in human-computer collaboration where each agent has different expertise.

### 2.2 Data collection

We created a schema containing 7 attributes and more than 3K entities (attribute values) in total. To elicit linguistic and strategic variants, we generate a random scenario for each task by varying the number of items, the number attributes, and the distribution of values for each attribute. See Appendix B for details of scenario generation.

We crowd-sourced dialogues on AMT by randomly pairing up users to perform the task within 5 minutes.[2] To discourage random guessing, we prevent users from selecting more than once every 10 seconds. The chat interface is shown in Appendix C. Our task was very popular and we collected 11K dialogues over a period of 13.5 hours.[3]

### 2.3 Dataset statistics

We show the basic statistics of our dataset in Table 3. An utterance is defined as a message sent by the user. The average utterance length is shorter due to the informality of the chat, however, a user usually sends multiple utterances in one turn. Some example dialogues are shown in Table 6 and Appendix H.

We coarsely categorize utterances into inform, ask, answer, greeting, apology by pattern matching (Appendix E). There are 7.4% multi-type utterances and 30.9% multi-entity utterances. In Ta-

---

[1]The dataset is available publicly at http://anonymous.

[2]If users exceed the time limit, the dialogue is marked as unsuccessful (but still logged).

[3]Tasks are put up in batches; the total time excludes intervals between batches.

[4]Entity names are replaced by entity types.

| Type | % | Standard example | Hard example |
|---|---|---|---|
| Inform | 30.4 | I know a judy. / I have someone who studied the bible in the afternoon. | **About equal** indoor and outdoor friends / **me too**. his major is forestry / **might be** kelly |
| Ask | 17.7 | Do any of them like Poi? / What does your henry do? | What can you tell me about our friend? / **Or maybe** north park college? |
| Answer | 7.4 | None of mine did / Yup / They do. / Same here. | yes 3 of them / No he likes poi / yes if boston college |

Table 1: Main utterance types and examples. We show both standard utterances whose meaning can be represented by typical logical forms (e.g. ask(indoor)), and open-ended ones which require much more complex logical forms (difficult parts in bold). Text spans corresponding to an entity are underlined.

| Phenomenon | Example |
|---|---|
| Coreference | (I know one Debra) does she like the indoors? / (I have two friends named TIffany) at World airways? |
| Coordination | keep on going with the fashion / Ok. let's try something else. / go by hobby / great. select him. thanks! |
| Chit-chat | Yes, that is **good ole** Terry. / All indoorsers! **my friends hate nature** |
| Categorization | same, most of mine are **female too / Does any of them names start with B** |
| Correction | I know one friend into Embroidery - her name is Emily. **Sorry – Embroidery friend is named Michelle** |

Table 2: Rich communication phenomena in the dataset. Evident parts is in bold and text spans corresponding to an entity are underlined. For coreference, the antecedent is in parentheses.

| | |
|---|---|
| # dialogues | 11439 |
| # completed dialogues | 9041 |
| Vocabulary size | 4548 |
| Average # of utterances | 11.41 |
| Average time taken per task (sec.) | 91.18 |
| Average utterance length (tokens) | 4.84 |
| Number of linguistic templates[4] | 30370 |

Table 3: Statistics of the MutualFriends dataset.

ble 1, we show example utterances where the rich semantics cannot be sufficiently represented by traditional slot-value pairs. Even the more standard ones can be challenging due to coreference and conjunctions.

Our dataset also exhibits interesting communication phenomena that add to its complexity. Coreference occurs frequently when people check attributes jointly. Sometimes the mentions are dropped: it simply continues from the previous partner utterance. People occasionally use external knowledge to group items with out-of-schema attributes (e.g., gender based on names, location based on schools). We summarize these phenomena in Table 2. In addition, we find 30% utterances involve cross-talk where the conversation does not progress linearly (e.g., italic utterances in Figure 1). Cross-talk is a common characteristic of online chat that creates ambiguity (Ivanovic, 2005).

One strategic aspect of this task is the order in which attributes are mentioned. We find that peo-

ple tend to start from attributes with fewer unique values, e.g., "all my friends like morning" given the $KB_B$ in Table 6, as intuitively it would help exclude items quickly given fewer values to check.[5] We provide a more detailed analysis of strategy in Section 4.2 and Appendix F.

## 3 Dynamic Knowledge Graph Network

The diverse semantics in our data motivates us to combine rich but unstructured representation of the dialogue history with structured knowledge. Our model consists of three components shown in Figure 2: (i) a dynamic knowledge graph, which represents the private KB and dialogue history in a graph (Section 3.1), (ii) a graph embedding over the nodes (Section 3.2), and (iii) an utterance generator (Section 3.3).

The knowledge graph represents entities and relations in the agent's private KB, e.g., google belongs to company. As the conversation unrolls, utterances are embedded and incorporated to node embeddings of mentioned entities. In Figure 2, "Most of my friends work for Google" updates the embedding of google. Further, mentioned nodes pass on this new information to their neighbors so that related entities (e.g., those in the same row or column) also receive the information. In our example, jessica and josh both receive new context when google and columbia are mentioned.

---

[5]Our goal is to model human behavior thus we do not discuss the optimal strategy here.

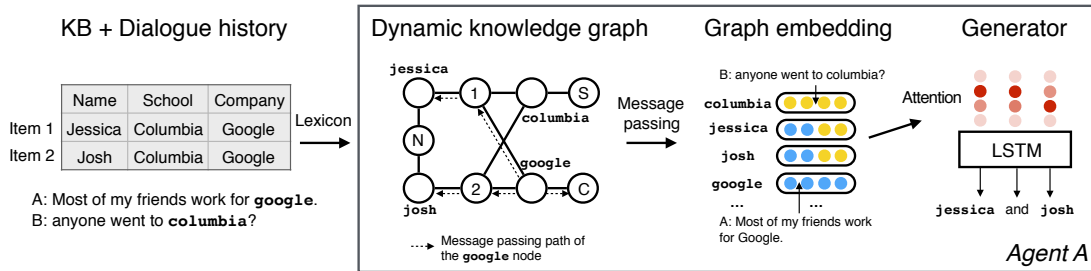

Figure 2: Overview of our approach. First, the KB and dialogue history (entities in **bold**) is mapped to a graph. Here, an item node is labeled by the item ID and an attribute node is labeled by the attribute's initial. Next, each node is embedded using relevant utterances through message passing. Finally, an LSTM generates the next utterance based on attention over the node embeddings.

The utterance generator, an LSTM, then produces the next utterance by attending to the node embeddings that represent the dialogue state.

### 3.1 Knowledge Graph

Given a dialogue of $T$ utterances, we construct an evolving graph $G_t$ over the KB and dialogue history for agent A.[6] We consider three types of nodes: item nodes, attribute nodes, and entity nodes. Edges between nodes represent their relations. For example, (item-1, hasSchool, columbia) means that the first item has attribute school whose value is columbia. An example graph is shown in Figure 2. The graph is updated at the end of an utterance if new entities (not in $KB_A$) are mentioned; they are added as dangling nodes.[7]

### 3.2 Graph Embedding

Given a knowledge graph, we are interested in computing a vector representation for each node $v$ that captures both its unstructured context from the dialogue history and its structured context in the KB. The node embedding $V$ is built from three parts: structural properties of an entity defined by the KB, embeddings of utterances in the dialogue history, and message passing between neighboring nodes.

**Node Features.** Simple structural properties of the KB often govern what is talked about; e.g., a high-frequency entity is often interesting (consider

---

6 It is important to differentiate perspectives of the two agents as they have different KBs. Thereafter we assume the perspective of agent A, i.e., accessing $KB_A$ and generating utterances for A only, and refer to B as the partner.

7 We use a rule-based lexicon to link text spans to entities. See details in Appendix D.

"All my friends like dancing."). We represent this type of information as an indicator vector $F(v)$, including the frequency and type (item, attribute, or entity type) of entity $v$.

**Mention Vectors.** A mention vector $M_t(v)$ contains unstructured context from utterances relevant to node $v$ up to turn $t$. Given embedding $u_t$ of the last utterance (Section 3.3), we define

$$M_t(v) = \lambda M_{t-1}(v) + (1 - \lambda)u_t,$$

$$\lambda = \begin{cases} \sigma\left(W^\lambda\left[M_{t-1}(v), u_t\right]\right) & \text{if } u_t \text{ mentions } v \\ 1 & \text{otherwise} \end{cases}$$

where $\sigma$ is the sigmoid function and $W^\lambda$ is a projection matrix. If no entity is mentioned in $u_t$, we update $v$ in $u_{t-1}$. This is useful when $u_t$ answers a question, e.g., "do you have any google friends?" "No." To differentiate between the agent's and the partner's utterances, $M_t(v)$ consists of two separate components for utterances from each source.

**Recursive Node Embeddings.** We propagate information between nodes according to the structure of the knowledge graph. In Figure 2, given "anyone went to columbia?", the agent should focus on her Google friends who went to Columbia. Therefore, we want this information to be sent to item nodes connected to columbia, and one step further to other attributes of these items because they might be implicitly referred to (google which was mentioned previously) or mentioned next (jessica and josh which will be generated).

We compute the embeddings recursively, analogous to belief propagation:

$$V_t^k(v) = \max_{v' \in N(v)} \tanh\left(W^m\left[V_t^{k-1}(v'), R(e_{v \to v'})\right]\right),$$

where $V_t^k(v)$ is the depth-$k$ node embedding at turn $t$, $W^m$ is a projection matrix, $R$ is an relation embedding function, $e_{v \rightarrow v'}$ denotes the relation from $v$ to $v'$, and max is an element-wise max operation.[8] Example message passing paths are shown in Figure 2.

The final node embedding is the concatenation of embeddings at each depth:

$$V_t(v) = \left[ V_t^0(v), \ldots, V_t^K(v) \right],$$

where $K$ is a hyperparameter (we experiment with $K \in \{0, 1, 2\}$) and $V_t^0(v) = [F(v), M_t(v)]$. The node embedding is updated periodically whenever $M_t(v)$ is updated by a new utterance.

### 3.3 Utterance Generation

In this section, we describe how an utterance is embedded and generated. Our model is based on a recurrent neural network (RNN). On turn $t$ (we elide the subscript $t$ for notational convenience), upon receiving an utterance of $n$ tokens, $\{x_n\}_{n=1}^N$, the RNN maps it to a vector: $h_n = \text{LSTM}_{\text{enc}}(h_{n-1}, x_n)$. We use the last hidden state $h_n$ as the utterance embedding $u$ which updates the mention vectors.

Given (updated) embeddings of each node, we use another RNN to generate the next utterance, where each token is either copied from a node or generated from the vocabulary. Formally,

$$h_n = \text{LSTM}_{\text{dec}}(h_{n-1}, [x_n, c_t]),$$

where $c_t$ is a weighted sum of node embeddings in the current turn: $c_t = \sum_i \alpha_{n,i} V_t(v_i)$. As shown in Figure 2, entities relevant to generating the next utterance should have high weights. We compute the weights through standard attention mechanism: $\alpha_n = \text{softmax}(a_n)$ and $a_{n,i} = w_a \tanh(W^a [h_{n-1}, V_t(v_i)])$, where $w_a$ is a scoring vector and $W^a$ is a projection matrix.

The RNN outputs a distribution over words in the vocabulary and the nodes in $G_t$ as in the copying mechanism of Jia and Liang (2016) :

$$p(x_{n+1} = w \mid x_{\leq n}, G_t) \propto \exp\left(U^w h_n^c + b\right),$$
$$p(x_{n+1} = r(v_i) \mid x_{\leq n}, G_t) \propto \exp\left(a_{n,i}\right),$$

where $w$ is a word in the vocabulary, $U^w$ and $b$ are the model parameters, and $r(v_i)$ is the realization of the entity represented by node $i$, e.g., generating "Google" given the node google.[9]

---

[8] Using sum slightly hurts performance.

[9] We realize an entity by sampling from the empirical distribution of its surface forms in the training set.

### 3.4 Entity Abstraction

In our setting, the role of an entity is governed by its relation to other entities and relevant utterances. For example, replacing google with alphabet in Figure 1 should make very little difference to the conversation. Note that we do not use an embedding matrix for any entity when computing $V(v)$. Furthermore, in the input embedding layer of the LSTM, we represent an entity by its type embedding concatenated with its node embedding. Accordingly, a regular word is represented as its word embedding concatenated with a zero vector of the same dimension as $V$. This way, the models' representation of an entity only depends on its structural property in the KB and the dialogue context.

## 4 Experiments

### 4.1 Setup

We use a one-layer LSTM with 100 hidden units and 100-dimensional word vectors for both the encoder and the decoder. We maximize the likelihood of each utterance from perspectives of both agents. The parameters are optimized by Ada-Grad (Duchi et al., 2010) with a learning rate of 0.5. We ran at least 10 epochs; after that, training stops if there is no improvement for 5 epochs. By default, we perform $K = 2$ iterations of message passing to compute node embeddings. We randomly split the data into train, dev, and test sets and only train on successful dialogues. For decoding, we sequentially sample from the output distribution with a softmax temperature of 0.5.[10] Hyperparameters are tuned on the dev set.

We compare DynoNet with its static cousion (StanoNet) and a rule-based system (Rule). StanoNet uses $G_0$ throughout the dialogue, thus the dialogue context is completely contained in the RNN hidden states instead of being structured around the knowledge graph. Rule maintains weights for each entity and each item in the KB to decide what to talk about and which item to select. It has a pattern-matching semantic parser, a rule-based policy, and a templated generator. See Appendix G for details.

---

[10] Since selection is a common 'utterance' in our dataset and neural generation models are susceptible to over-generating common sentences, we halve its probability during sampling.

## 4.2 Evaluation

We test our systems in two interactive settings: bot-bot chat and bot-human chat. We perform both automatic evaluation and human evaluation to test syntactic, semantic, and pragmatic competence of each system.

**Automatic Evaluation.** First, we compute the cross-entropy ($\ell$) of a model on test data. As shown in Table 4, DynoNet has the lowest test loss. Next, we have a model chat with itself on the scenarios from the test set.[11] We evaluate the chats with respect to language variation, effectiveness and strategy.

For language variation, we report the average utterance length $L_u$ and the unigram entropy $H$ in Table 4. Compared to Rule, the neural models tend to generate shorter utterances (Li et al., 2016b; Serban et al., 2017). However, they are more diverse; for example, questions are asked in multiple ways such as "Do you have ...", "Any friends like ...", "What about ...".

At the discourse level, we expect the distribution of a bot's utterance types to match the distribution of human's. We show percentages of each utterance type in Table 4. For Rule, the decision about which action to take is written in the rules, while StanoNet and DynoNet learned to behave in a more human-like way, frequently informing and asking questions.

To measure effectiveness, we compute the success rate per turn ($C_T$) and per selection ($C_S$). In Table 4, humans are the best at this game, followed by Rule which is comparable to DynoNet. Next, we investigate the strategies leading to these results.

An agent needs to decide which entity/attribute to check first to quickly reduce the search space. We hypothesize that humans tend to first focus on a majority entity and an attribute with fewer unique values (Section 2.3). We show the average frequency of first-mentioned entities (#$Ent_1$) and the average number of unique values for first-mentioned attributes ($|Attr_1|$) in Table 4.[12] Clearly, both DynoNet and StanoNet successfully matched human's starting strategy by favoring more frequent entities and attributes of smaller do-

main size.

To examine the overall strategy, we show the average number of attributes and entities mentioned during the conversation in Table 4. Humans and DynoNet strategically focus on a few attributes and entities, whereas Rule needs almost twice entities to achieve similar success rates. This suggests that the effectiveness of Rule mainly comes from large amounts of unselective information, which is consistent with feedback from their human partners.

**Partner Evaluation.** We generated 200 new scenarios and put up the bots on AMT using the same chat interface that used for data collection. Each AMT worker is randomly paired with Rule, StanoNet, DynoNet, or another human (but they don't know which), and we make sure that all four types of agents are tested in each scenario (800 dialogues in total). At the end of each task, humans are asked to rate their partner in terms of fluency, correctness, cooperation, and human-likeness from 1 (very bad) to 5 (very good), along with optional comments.

As shown in the example dialogues in Table 6, DynoNet cooperates smoothly with the human partner, e.g., replying with relevant information about morning/indoor friends when the partner mentioned that all her friends prefer morning and most like indoor. However, it "lied" when saying that it had a morning friend who likes outdoor. StanoNet starts well but doesn't continue with the morning friend, because the `morning` node is not updated dynamically when mentioned by the partner. Rule always informs true facts but poorly follows the partner. In the comments, most complaint about Rule is that it was not 'listening' or 'understanding'. We show the average ratings in Table 5 and the histograms in Appendix H. Overall, DynoNet achieves better partner satisfaction, especially in cooperation.

**Third-party Evaluation.** Since the evaluator cannot see the partner's KB, judgment on correctness is a mere guess. Therefore, we also created a *third-party evaluation* task, where an independent AMT worker is shown a conversation and the KB of one of the agents; she is asked to rate the same aspects of the agent as in the partner eval-

---

[11] We limit the number of turns in bot-bot chat to be the maximum number of turns humans took in the test set (46 turns).
[12] Both numbers are normalized to $[0, 1]$ with respect to all entities/attributes in the corresponding KB.

[13] We only show the most frequent speech acts therefore the numbers do not sum to 1.
[14] For third-party evaluation, we first take mean of each question then average the ratings.

| System | $\ell$ | $L_u$ | $H$ | Sel | Inf | Ask | Ans | Greet | $C$ | $C_T$ | $C_S$ | #Ent$_1$ | \|Attr$_1$\| | #Ent | #Attr |
|---|---|---|---|---|---|---|---|---|---|---|---|---|---|---|---|
| Human | - | 5.10 | 4.57 | .21 | .31 | .17 | .08 | .08 | .82 | .07 | .38 | .55 | .35 | 6.1 | 2.6 |
| Rule | - | 7.53 | 3.37 | .18 | **.34** | .23 | .00 | .12 | .90 | .05 | **.30** | .24 | .60 | 9.7 | 3.1 |
| StanoNet | 2.20 | 4.01 | **4.05** | .19 | .26 | .12 | .23 | **.09** | .78 | .04 | .18 | .61 | **.19** | 7.1 | 2.9 |
| DynoNet | **2.13** | 3.37 | 3.90 | **.22** | .26 | **.13** | .20 | .12 | **.96** | **.06** | .25 | **.55** | .18 | **5.2** | **2.5** |

Table 4: Automatic evaluation on human-human and bot-bot chats on test scenarios. Best results (except Human) are in bold. Neural models generate shorter (lower $L_u$) but more diverse (higher $H$) utterances. Overall, their distributions of utterance types match human's.[13] Rule is effective in completing the task, but it is not information-efficient given the large number of attributes (#Attr) and entities (#Ent) mentioned.

| System | $C$ | $C_T$ | $C_S$ | Partner eval | | | | Third-party eval | | | |
|---|---|---|---|---|---|---|---|---|---|---|---|
| | | | | Flnt | Crct | Coop | Human | Flnt | Crct | Coop | Human |
| Human | .77 | .06 | .34 | 4.2$^{rds}$ | 4.3$^{rds}$ | 4.2$^{rds}$ | 4.1$^{rds}$ | 4.0 | 4.3$^{ds}$ | 4.0$^{ds}$ | 4.1$^{rds}$ |
| Rule | **.85** | **.06** | **.29** | 3.6 | 4.0 | 3.5 | 3.4 | 4.0 | **4.4**$^{hds}$ | **3.9**$^s$ | **4.0**$^s$ |
| StanoNet | .73 | .04 | .23 | 3.5 | 3.8 | 3.4 | 3.3 | 4.0 | 4.0 | 3.8 | 3.8 |
| DynoNet | **.85** | .05 | .27 | **3.8**$^{rs}$ | 4.0 | **3.8**$^{rs}$ | **3.6**$^s$ | 4.0 | 4.1 | **3.9**$^s$ | 3.9 |

Table 5: Results on human-bot/human chats. Best results (except Human) in each column are in bold. We report the average ratings of each system.[14] DynoNet has the best partner satisfaction in terms of fluency (Flnt), correctness (Crct), cooperation (Coop), human likeness (Human). The superscript of a result indicates that its advantage over other systems ($r$: Rule, $s$: StanoNet, $d$: DynoNet) is statistically significant with $p < 0.05$ using paired $t$-tests.

uation and provide justifications.[15] The average ratings and histograms are shown in Table 5 and Appendix H respectively.

Surprisingly, there is a discrepancy between the two evaluation modes. Manual analysis of the comments indicates that while a participant in the dialogue considers the partner fluent as long as there is no communication overhead, a third person may have higher standards, e.g., thinking short sentences do not fully reveal an agent's fluency. For human-likeness, partner evaluation is largely correlated with coherence (e.g., not repeating or ignoring past information) and task success, whereas third-party evaluators often rely on informality (e.g., usage of colloquia like "hiya", capitalization, and abbreviation) or intuition.[16] Interestingly, third-party evaluators noted most phenomena listed in Table 2 as indicators of human-beings, e.g., correcting oneself, making chit-chat other than simply finishing the task.

### 4.3 Ablation Studies

Our model has two novel designs: entity abstraction and message passing for node embeddings.

---
[15] Each agent in a dialogue is rated by at least 5 people.
[16] Nevertheless, informality can be easily handcrafted or learned.

Table 7 shows what happens if we ablate these. When the number of message passing iterations, $K$, is reduced from 2 to 0, the loss consistently increases. Removing entity abstraction—meaning adding entity embeddings to node embeddings and the LSTM input embeddings—also degrades performance. This shows that DynoNet benefits from contextually-defined, structural node embeddings rather than ones based on a classic lookup table.

| Model | $\ell$ |
|---|---|
| DynoNet (K = 2) | **2.16** |
| DynoNet (K = 1) | 2.20 |
| DynoNet (K = 0) | 2.26 |
| DynoNet - entity abstraction | 2.21 |

Table 7: Ablations of our model on the dev set.

## 5 Discussion

There has been a recent surge of interest in end-to-end task-oriented dialogue systems. Unfortunately, the progress has been limited by the size of available datasets (Serban et al., 2015a). Most work uses a small corpus (< 5K dialogues) through Wizard-of-Oz data collection (Williams et al., 2016; Asri et al., 2016) or simulators (Bordes and Weston, 2017; Li et al., 2016d), which

| Friends of A | | | | |
|---|---|---|---|---|
| ID | Name | Company | Time | Location |
| 1 | Kathy | TRT Holdings | afternoon | indoor |
| 2 | Jason | Dollar General | afternoon | indoor |
| 3 | Johnny | TRT Holdings | afternoon | outdoor |
| 4 | Frank | SFN Group | afternoon | indoor |
| 5 | Catherine | Dollar General | afternoon | indoor |
| 6 | Catherine | Weis Markets | afternoon | indoor |
| 7 | Kathleen | TRT Holdings | morning | indoor |
| 8 | Lori | TRT Holdings | afternoon | indoor |
| 9 | Frank | L&L Hawaiian Barbecue | afternoon | outdoor |

| Friends of B | | | | |
|---|---|---|---|---|
| ID | Name | Company | Time | Location |
| 1 | Justin | New Era Tickets | morning | indoor |
| 2 | Kathleen | TRT Holdings | morning | indoor |
| 3 | Gloria | L&L Hawaiian Barbecue | morning | indoor |
| 4 | Kathleen | Advance Auto Parts | morning | outdoor |
| 5 | Justin | Arctic Cat | morning | indoor |
| 6 | Anna | Dollar General | morning | indoor |
| 7 | Steven | SFN Group | morning | indoor |
| 8 | Wayne | R.J. Corman Railroad Group | morning | indoor |
| 9 | Alexander | R.J. Corman Railroad Group | morning | indoor |

**A: Human    B: Human**

A: Hi
B: hey || i have one outdoor
A: I have 4 TRT Holdings || I have 2 outdoor one Johnny, other Frank
B: i only have one TRT Holdings - Kathleen
A: SELECT 7    B: SELECT 2

**A: DynoNet    B: Human**

A: **hi**
B: Hello || all my friends prefer morning
A: **1 of my morning likes the outdoors**
B: and all like indoor except for one
A: **do they work for trt holdings?**
B: Kathleen?
A: **SELECT 7**    B: SELECT 2

**A: StanoNet    B: Human**

A: **Hello**
B: hi
A: **Hello || I have one morning person.**
B: all of my friends like mornings
A: **My friend prefers afternoon works at trt holdings.**
B: what is their name?
A: **Likes indoors.**
B: what is your fiend who likes morning name?
A: **They work for trt holdings.**
B: SELECT 2    A: **SELECT 7**

**A: Human    B: Rule**

B: **hiya**
A: hEY
B: **I have 1 indoors and kathleen.**
A: Most of mine are indoors.
B: **SELECT 1**
A: I have one morning and rest afternoon.
B: **Do you have any friend working at l hawaiian?**
A: I don't know Justin
B: **I have 1 alexander.**
...

Table 6: Examples human-bot chats. The mutual friend is highlighted in blue in each KB. Bots' utterances are in bold and selected items are represented by item IDs. Only the first half of the human-Rule chat is shown due to space limit. Multiple utterances of one agent is separated by ||.

mainly focuses on information-query tasks. Current strategic dialogue datasets include Settlers of Catan (Afantenos et al., 2012) (2K turns) and the cards corpus (Potts, 2012) (1.3K dialogues), though neither has offered an end-to-end dialogue system.

Most task-oriented dialogue systems follow the POMDP-based approach (Young et al., 2013). Despite their success (Wen et al., 2017; Dhingra et al., 2016), the requirement for handcrafted slots limits their scalability to new domains and burdens data collection with extra state labeling. To break this limit, Bordes and Weston (2017) propose a Memory-Networks-based approach without domain-specific features. However, the memory is unstructured and interfacing with KBs relies on API calls, whereas our model embeds both the dialogue history and the KB structurally. Williams and Zweig (2016) use an LSTM to automatically infer the dialogue state, but as they focus on dialogue control rather than the full prob-

lem, the LSTM input contains only recognized entities, which restricts the representational power for richer utterances. Our network architecture is most similar to EntNet (Henaff et al., 2017), where memories are also updated by input sentences recurrently. The main difference is that our model allows information to be propagated between structured entities, which is shown to be crucial in our setting (Section 4.3).

In conclusion, we believe the symmetric collaborative dialogue setting and our dataset provide unique opportunities at the interface of traditional task-oriented dialogue and open-domain chat. We also offered DynoNet as a promising means for open-ended dialogue state representation. Looking forward, our dataset facilitates the study of pragmatics and human strategies in dialogue—a good stepping stone towards learning more complex dialogues such as negotiation. It would be interesting to integrate supervised learning with game-theoretic approaches in future.

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
