# Peer review of "Learning Symmetric Collaborative Dialogue Agents with Dynamic Knowledge Graph Embeddings"

_ACL 2017 — decision unknown_

[Official Review · Reviewer 1 · rating 3 · confidence 3]
soundness 3 · originality 4 · clarity 3 · impact 3 · substance 3 · appropriateness 5 · meaningful comparison 4 · presentation format Oral Presentation

This paper proposes a method for building dialogue agents involved in a
symmetric collaborative task, in which the agents need to strategically
communicate to achieve a common goal.  

I do like this paper.  I am very interested in how much data-driven techniques
can be used for dialogue management.  However, I am concerned that the approach
that this paper proposes, is actually not specific to symmetric collaborative
tasks, but to tasks that can be represented as graph operations, such as
finding an intersection between objects that the two people know about.

In Section 2.1, the authors introduce symmetric collaborative dialogue setting.
 However, such dialogs have been studied before, such as Clark and Wilkes-Gibbs
explored (Cognition '86), and Walker's furniture layout task (Journal of
Artificial Research '00).

On line 229, the authors say that this domain is too rich for slot-value
semantics.  However, their domain is based on attribute value pairs, so their
domain could use a semantics represenation based on attribute value-pairs, such
as first order logic.

Section 3.2 is hard to follow.        The authors often refer to Figure 2, but I
didn't find this example that helpful.        For example, for section 3.1, at what
point of the dialogue does this represent?  Is this the same after `anyone went
to columbia?'

[Official Review · Reviewer 2 · rating 3 · confidence 4]
soundness 3 · originality 4 · clarity 3 · impact 3 · substance 4 · appropriateness 5 · meaningful comparison 4 · presentation format Poster

This paper presents a novel framework for modelling symmetric collaborative
dialogue agents by dynamically extending knowledge graphs embeddings. The task
is rather simple: two dialogue agents (bot-bot, human-human or human-bot) talk
about their mutual friends. There is an underlying knowledge base for each
party in the dialogue and an associated knowledge graph. Items in the knowledge
graph have embeddings that are dynamically updated during the conversation and
used to generate the answers.

- Strengths: This model is very novel for both goal-directed and open ended
dialogue. The presented evaluation metrics show clear advantage for the
presented model.

- Weaknesses: In terms of the presentation, mathematical details of how the
embeddings are computed are not sufficiently clear. While the authors have done
an extensive evaluation, they haven't actually compared the system with an
RL-based dialogue manager which is current state-of-the-art in goal-oriented
systems. Finally, it is not clear how this approach scales to more complex
problems. The authors say that the KB is 3K, but actually what the agent
operates is about 10 (judging from Table 6).

- General Discussion: Overall, I think this is a good paper. Had the
theoretical aspects of the paper been better presented I would give this paper
an accept.